# Tidal Wave-Driven Variability in the Mars Ionosphere-Thermosphere System

**Scott A. Thaller** [1,*], **Laila Andersson** [1], **Marcin Dominik Pilinski** [1], **Edward Thiemann** [1], **Paul Withers** [2], **Meredith Elrod** [3,4], **Xiaohua Fang** [1], **Francisco González-Galindo** [5], **Stephen Bougher** [6] **and Geoffrey Jenkins** [6]

1   Laboratory for Atmospheric and Space Physics, University of Colorado Boulder, Boulder, CO 80303, USA;
    Laila.Andersson@lasp.colorado.edu (L.A.); Marcin.Pilinski@lasp.colorado.edu (M.D.P.);
    Ed.Thiemann@lasp.colorado.edu (E.T.); Xiaohua.Fang@lasp.colorado.edu (X.F.)
2   Center for Space Physics, Boston University, Boston, MA 02215, USA; withers@bu.edu
3   NASA Goddard Spaceflight Center, Greenbelt, MD 20771, USA; meredith.k.elrod@nasa.gov
4   Center for Research and Exploration in Space Science & Technology (CRESST), University of Maryland,
    College Park, MD 20742, USA
5   Instituto de Astrofísica de Andalucía-Consejo Superior de Investigaciones Científicas (IAA-CSIC),
    18008 Granada, Spain; ggalindo@iaa.es
6   Climate and Space Sciences and Engineering Department, University of Michigan,
    Ann Arbor, MI 48109, USA; bougher@umich.edu (S.B.); geoffj@umich.edu (G.J.)
*   Correspondence: scott.thaller@lasp.colorado.edu

**Abstract:** In order to further evaluate the behavior of ionospheric variations at Mars, we investigate the Martian ionosphere-thermosphere (IT) perturbations associated with non-migrating thermal tides using over four years of Mars Atmosphere and Volatile Evolution (MAVEN) in situ measurements of the IT electron and neutral densities. The results are consistent with those of previous studies, namely strong correlation between the tidal perturbations in electron and neutral densities on the dayside at altitudes ~150–185 km, as expected from photochemical theory. In addition, there are intervals during which this correlation extends to higher altitudes, up to ~270 km, where diffusive transport of plasma plays a dominant role over photochemical processes. This is significant because at these altitudes the thermosphere and ionosphere are only weakly coupled through collisions. The identified non-migrating tidal wave variations in the neutral thermosphere are predominantly wave-1, wave-2, and wave-3. Wave-1 is often the dominant wavenumber for electron density tidal variations, particularly at high altitudes over crustal fields. The Mars Climate Database (MCD) neutral densities (below 300 km along the MAVEN orbit) shows clear tidal variations which are predominantly wave-2 and wave-3, and have similar wave amplitudes to those observed.

**Keywords:** Mars ionosphere; Mars thermosphere; tidal waves; ionosphere-thermosphere coupling

## 1. Introduction

An important source of variability in the Martian ionosphere is that because of solar thermal tides in the thermosphere [1–5]. Thermospheric and ionospheric tides are observable in situ as perturbations in the neutral atmospheric and electron densities, respectively, as a function of longitude, typically over a limited range of local time, solar zenith angle, and latitude [1,6] (see Appendix A). The variations in the neutral atmosphere are driven by an interaction of solar heating and longitudinal/zonal asymmetries of the surface topography [7–12]. These tidal perturbations in the neutral atmosphere can give rise to variations in the ionosphere. At a given altitude, electrons and ions are produced at a rate dependent on the local EUV irradiance and neutral density. The resulting photo-produced density will increase until

the recombination rate between the electrons and ions balances production. Photochemical equilibrium (see Appendix B) describes ionosphere-thermosphere coupling dynamics under the condition that the timescale of dissociative recombination, and hence the production rate at equilibrium, is smaller than the timescales of diffusive processes, this typically occurs below ~200 km [5,13]. At photochemically controlled altitudes (<200 km) the perturbations in the neutral thermosphere density drive those in the ionospheric electron density [1,4,14]. Bougher et al. [1,14] showed that the altitude at which the peak electron density occurs (typically ~134 km near the subsolar point), varies in phase with the neutral atmosphere density modulations, a behavior consistent with photochemical equilibrium theory. Mendillo et al. [4] studied the electron density at a similar, but constant, altitude (135 km), and showed the tidal variation in the ionosphere to be associated with those in the neutral thermosphere, which in turn were driven by wave-2 tidal waves. By including the relevant parameters of EUV flux, neutral density, electron temperature, etc., they showed the variations to take place according to photochemical equilibrium. Fang et al., [15] have used a numerical model to predict that electron density profile responds to changes in neutral density caused by thermospheric heating during dust storms. They found that the electron profile was characterized by a Chapman profile below ~200 km, but that at higher altitudes plasma processes where also involved. Mayyasi et al. [16] compared dayside ion and neutral measured densities in the Martian ionosphere-thermosphere system between ~150 km and 200 km and showed the plasma and neutral atmosphere to be strongly correlated in 70% of the density-altitude profiles studied.

In order to study the variability in the Martian ionosphere associated with non-migrating tides we sampled over two Mars' years of ionosphere-thermosphere tidal wave correlation coefficients for observations between periapsis and 300 km. We show that there are intervals during which the tidal variations in the thermosphere and ionosphere are well correlated at altitudes (>200 km) above the photochemistry-controlled region. Since at these altitudes the ionosphere and thermosphere are only weakly coupled through ion-neutral collisions (i.e., the collisional rates are low) the most likely explanation of this phenomenon is that while the upper (>200 km) and lower (<200 km) thermospheric regions are varying in similar fashion to one another, the (photochemical-controlled) electron density at low altitudes is directly controlling the high (non-photochemical) altitudes, via being the source for the vertically transported electrons. We also evaluated comparisons of the neutral wave density variations measured by MAVEN to the Mars Climate Database (MCD) model results and found that the model captures the mean wave amplitudes and wave numbers accurately.

## 2. Data Processing and Methodology

The data sets used in this study are in situ Mars' ionospheric electron densities and thermospheric neutral densities made respectively by the Langmuir Probe and Waves (LPW) [17] and the Neutral Gas and Ion Mass Spectrometer (NGIMS) [18] instruments onboard the Mars Atmosphere and Volatile Evolution (MAVEN) spacecraft [19]. We also use model data from the Mars Climate Database (MCD) [20,21], version 5.3, taken from times and location corresponding to those of a virtual MAVEN satellite flying through the model space. The total datasets used extend from 1 February 2015 through 1 July 2019. The LPW measured electron densities and NGIMS closed source neutral carbon dioxide ($CO_2$), oxygen (O), diatomic nitrogen ($N_2$), carbon monoxide (CO), and argon (Ar) abundances are selected along the inbound trajectory of MAVEN's orbit from 300 km to periapsis (typically ~150 km, and ~130 km for deep dip and aerobraking campaigns) over ~twenty-day intervals and sorted by altitude and longitude. Only the inbound segments are used in order to avoid effects of adsorption of gas on the NGIMS spectrometer cell wall [16,22]. NGIMS argon coverage routinely extends to higher altitudes than do the other neutral species; so, we use both neutral argon density and calculated neutral mass density. The neutral mass density, $n_{mass}$, of the Martian atmosphere is approximated from the $CO_2$, O, $N_2$, and CO abundances as $n_{mass} = n_{CO2}m_{CO2} + n_O m_O + n_{N2}m_{N2} + n_{CO}m_{CO}$ where $n_x$ and $m_x$ are number density and mass, respectively, of neutral species x. The twenty-day window is advanced in 10-day intervals so that consecutive 20-day intervals overlap by 10 days. The sorting of

the MAVEN data and MCD results are illustrated in Figure 1. In each 20-day window, the arithmetic mean of the electron, argon, and neutral mass densities are determined as a function of altitude and longitude, per 10 km × 15° bins; this is shown in Figure 1a. To verify the appropriateness of this resolution, experiments with bins of 5 km resulted in the same results for this study. The mean densities in each 10 km altitude increment between 300 km and periapsis independent of longitude, i.e., bins of 10 km × 360°, are determined (Figure 1) and subtracted from the mean density per 10 km × 15° bin at the corresponding altitudes. The associated percent difference in each altitude-longitude bin from the mean density for that altitude are calculated by dividing this difference by the mean density in the 10 km × 360° bin (Figure 1c).

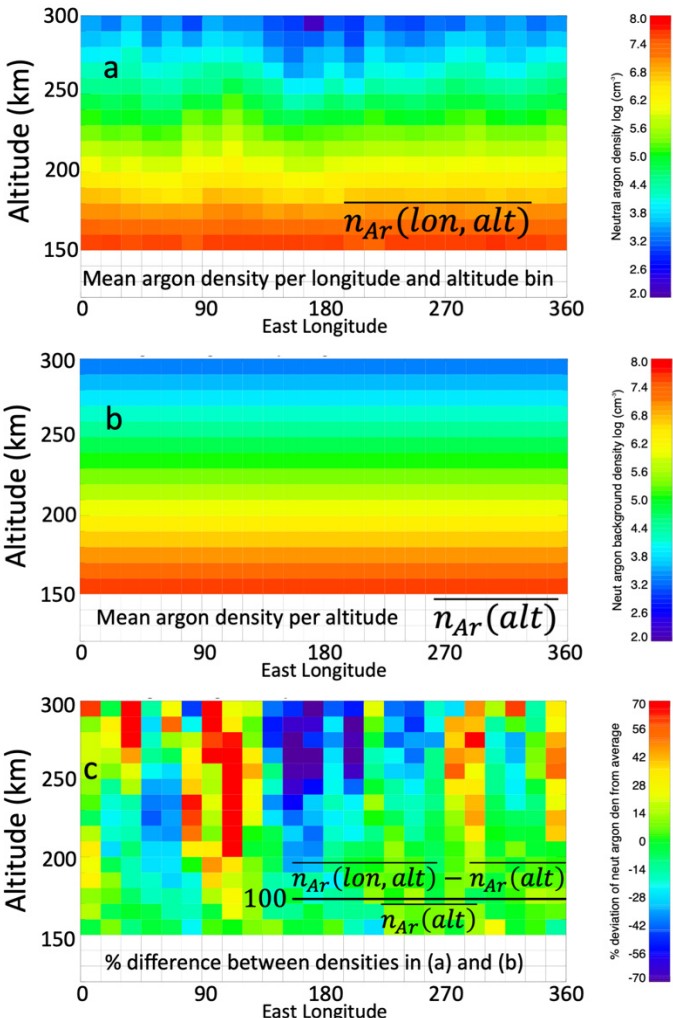

**Figure 1.** Illustration of tidal wave perturbation extraction with an example using Mars Atmosphere and Volatile Evolution (MAVEN) neutral argon number densities from 26 March–14 April 2016. Panel (**a**) shows the mean neutral argon number densities in bins of 10 km by 15° longitude. Panel (**b**) shows the mean neutral argon number densities in altitude for all longitudes. Panel (**c**) shows the percent difference between the mean argon densities in (**a**) and (**b**).

The electron and neutral density measurements are made as MAVEN passes though the upper atmosphere of Mars at different longitudes every orbit. Prior to the aerobraking campaign from February to April 2019, MAVEN had an orbital period of ~4.5 h. Mars' rotation period is 24.62 h, so the longitude of each MAVEN orbit periapsis has moved ~65.8° from the previous orbit. Because this is not an integer multiple of 360°, the longitudes visited at the periapsis passes are different. It takes about ~14 days to accumulate a longitude coverage with an average separation of less than 10°.

Further, the precession of the orbit results in a slow drift of the periapsis in local time, roughly ~1–5 h, in ~20 days depending on the orbit orientation. The result is, for MAVEN, that an approximately full coverage in longitude can be made over a relatively localized local time sector in ~20 days.

Modeled neutral densities and electron densities are taken from the MCD version 5.3 for times and locations corresponding to the MAVEN trajectory; effectively flying a virtual MAVEN through the simulation data. The MCD is a database of mean Mars atmospheric values derived from the Laboratoire de Météorologie Dynamique (LMD) Mars Global Climate Model (GCM) simulations. The MCD (version 5.3) consists of the LMD-GCM model outputs with grid resolution of $5.625° \times 3.75°$ longitude-latitude 12 times per day (2-h intervals) for a "mean day" every Mars' month defined as Solar Longitude ($L_s$) intervals of $30°$. This captures the main seasonal and diurnal variations. The average Mars atmosphere for particular dates and locations is inferred from interpolation of the database. Post-processing procedures of the MCD results with high-resolution MOLA topography and atmospheric mass corrections allow for a higher resolution of 32 pixels per degree. The post-processing routines run on the database by the MCD software [21] also include options to implement various dust and EUV flux scenarios as well as small scale (i.e., gravity waves) and large-scale atmospheric perturbations. A realistic spatial and temporal variability of the large-scale atmospheric variations are stored in Empirical Orthogonal Functions (EOFs), see Forget et al. [21] for details. The MCD model includes photochemistry; the associated electron densities are calculated only when the pressure is $5 \times 10^{-6}$ Pa or greater, which is typically below ~200 km. The MCD runs used in this study use the high resolutions mode (32 pixels /degree), the "climatology" dust scenario which is designed to be representative of a baseline for a typical Martian year, and average conditions for solar EUV (E10.7) flux. The MCD average EUV condition corresponds to a E10.7 value at earth of $140 \times 10^{-22}$ W/m$^2$/Hz [21]. The MCD runs for this study also include both small and large-scale perturbations.

The density perturbations associated with the tidal waves are analyzed as follows. The percent density variations as a function of longitude for each 10 km altitude range, such as those illustrated in Figure 1c, are fitted with a wave-4 harmonic model. The harmonic model [12], uses a least-squares fit set to include wave-0 through wave-4 harmonics. We have also conducted test runs using harmonics up to 5. These modifications had very little impact on the results of this study; with the wave amplitudes and correlations essentially unchanged. Wave-5 are sometimes present, but mainly in and around those locations where the wave-0 to wave-4 model indicates wave-4. These fits yield the amplitudes and phases for the various wave number components of the density perturbations. The amplitudes of individual fitted wave components returned are compared and the harmonic having the largest amplitude relative to the others is selected as the dominant wavenumber. To quantify how well various pairs of wave perturbations are phased, we calculate the Pearson's correlation coefficients between the wave fits for the tidal variations in electron density and the neutral densities (both the argon number density and neutral mass density), as well as between the MAVEN and MCD neutral mass densities.

Examination of neutral-electron correlation is important. At photochemically controlled altitudes, above the peak electron density, there should be a positive correlation between the electron density and neutral density [1,4] (see Appendix B). The MAVEN-MCD neutral correlations are important for evaluation of how often and where MCD model reproduces the MAVEN observations.

Evaluation of the dataset is done separately for the night and day because the photo-ionization is occurring on the dayside. Statistics on the average tidal wave amplitude and correlation coefficient between the electron and neutral density variations in altitude are determined for the night (19–5 solar local time (SLT)) and day (7–19 SLT) side wave fit profiles separately (excluding the dawn terminator, 5–7 SLT, where the thermosphere exhibits more complex behavior, e.g., larger amplitude gravity waves [23] and the ionosphere exhibits plasma-specific structures [16]) and finding the mean absolute wave amplitude in each altitude range, of the full wave fit amplitude (over all longitudes). Experiments with calculating the mean RMS wave amplitude per altitude gave similar amplitudes and curve shape in amplitude-altitude space. Similarly, the average correlation coefficient per altitude on

the day and night sides are determined as the mean value of the correlation coefficients between the electron density and neutral mass density wave fits, per altitude range, on the night and day sides.

## 3. Example Cases

In this section we use two example case intervals to illustrate the observed tidal variations and IT tidal correlations. Figure 2 shows an example of the tidal mode variations in the neutral mass density of Mars' atmosphere (Figure 2a,b) and ionospheric electron density (Figure 2c) and in the MCD neutral atmosphere model results (Figure 2d). This interval is a subset of a longer interval (10 March–16 April 2016) studied by England et al. [24] for a study in which they focused on the tidal waves in the neutral atmospheric constituents. England et al. [24] found that the dominant wavenumber during this interval was wave-3, which is consistent with the results in this study. During the interval in Figure 2, MAVEN's periapsis is located in the northern hemisphere and around ~71–74° north latitude, the local time of the coverage moves from ~18 to ~13 h, solar zenith angle (SZA) from ~72° to ~55° and $L_s$ from ~125° to ~138°, i.e., late Martian northern summer/early fall.

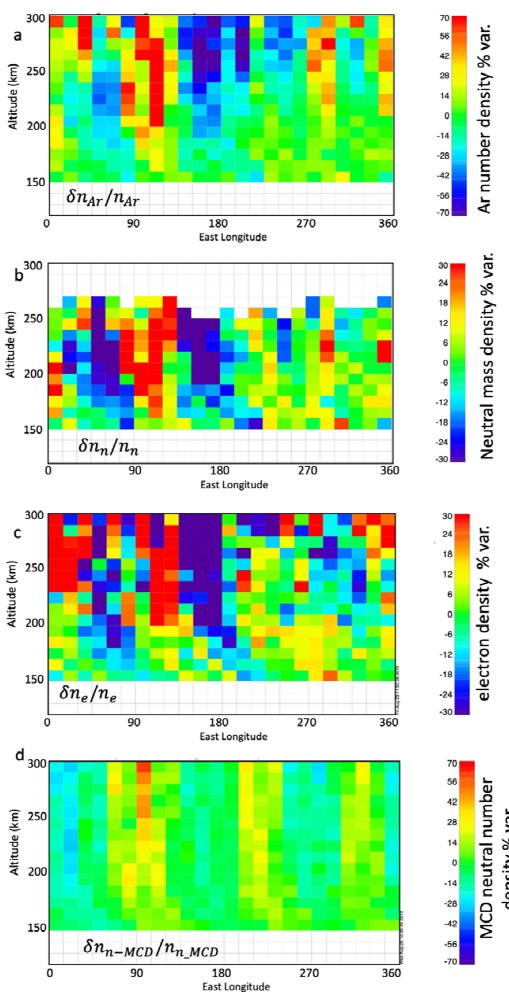

**Figure 2.** The percent density variations based on 20 days in March and April 2016; solar local time (SLT) ~18–13 h, solar zenith angle (SZA) ~72–55°, LAT ~71–74° north, and $L_s$ from ~125° to ~138° MAVEN argon density percent variation (**a**), neutral mass density percent variation (**b**), electron density percent variation (**c**), and Mars Climate Database (MCD) model neutral mass density percent variation (**d**). Note the color scale for neutral argon and MCD densities goes from −70% to 70%, and that for the neutral mass and electrons from −30% to 30%.

Comparing panels 2a, 2b, and 2c, at lower altitudes (150–200 km), density enhancements can be seen at longitudes ~30°, 150°, and 290° in all three quantities. Above ~200 km, the location of the wave crests shifts westward by about ~15° and the normalized density perturbations become larger. These features can also be seen by comparing the waveforms presented in Figure 3 with those in Figure 4.

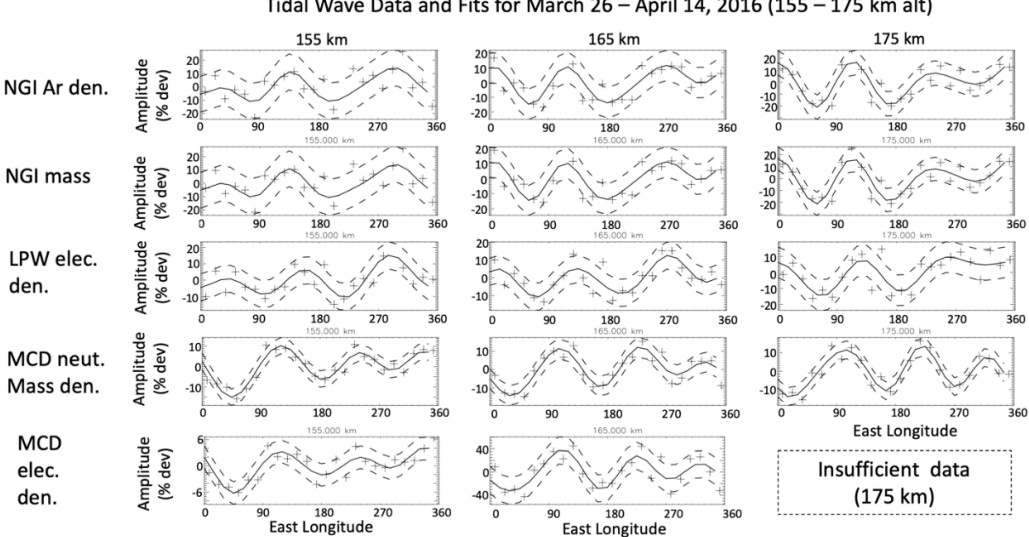

**Figure 3.** Tidal variations (+) taken from 155 km, 165 km, and 175 km in Figure 2, corresponding 4th order (wave-0 to wave-4) fits (solid line) and 1-sigma fit errors (dashed lines), in the neutral argon density (top row), neutral mass density (second row from top), ionospheric electron density (third row), MCD neutral mass density (fourth row) and MCD electron density (bottom row) for 20 days from 26 March to 14 April 2016. SLT ~18–13 h., SZA ~72–55°, LAT ~71–74° north, and $L_s$ from ~125° to ~138°.

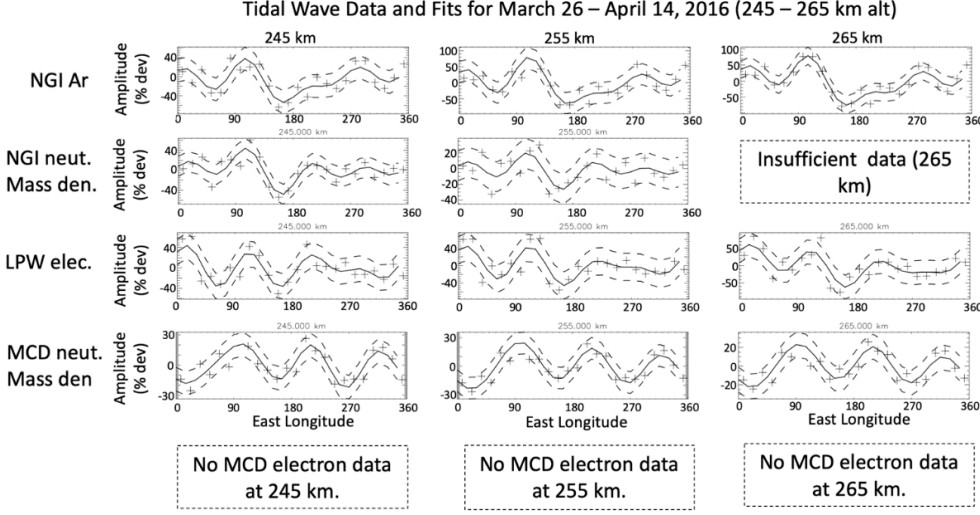

**Figure 4.** Tidal waves and corresponding fits (from the top row down) for the neutral argon density, neutral mass density, electron density, and MCD model density, for 26 March–14 April 2016 for higher altitudes, 245 km to 265 km, SLT ~18–13 h, SZA ~72–55°, LAT ~71–74° north, and $L_s$ from ~125° to ~138°.

Figure 3 shows individual plots of the neutral and electron density variation at altitudes 155–175 km for the same data shown in Figure 2. The "+" symbols are the actual data, the solid black line is the fit using the wave-0 to wave-4 decomposition described above, and the dashed lines are the 1 sigma uncertainties. The results of the wave analysis in Figure 3 present the tidal variation at 155 km, 165 km, and 175 km, in the neutral argon mass density (top row), neutral mass density (second row from top), ionospheric electron density (third row), MCD neutral mass density (fourth row), and MCD electron

density (bottom row). Both the MAVEN and MCD tidal waves are dominated primarily by wave-3 oscillations in this example.

At these altitudes (155–175 km), on the dayside, a positive correlation between electrons and neutral densities is consistent with tidal variation in the ionosphere driven by variation in the neutral atmosphere in photochemical equilibrium with the ionosphere. The amplitudes of the electron density perturbations are typically between a factor of ~0.5 and ~1 of the neutral amplitudes. A factor of 0.5 is expected for an environment controlled only by photochemical equilibrium (see Appendix B), a larger ratio may imply diffusive transport is also playing a significant role. The amplitudes of the MCD electron tidal perturbations at ~155 km are about $\frac{1}{2}$ those of the MCD neutrals, but at higher altitudes, ~165 km, the electron tidal waves are larger almost by a factor of 3. The data also show a phase difference between the MCD tidal perturbation and those observed by MAVEN, by about ~50° in Figure 3; the MAVEN and MCD wave amplitudes are similar.

Figure 4 shows the same kind of wave analysis for higher altitudes, 245 km, 255 km, and 265 km; the rows are ordered the same as in Figure 3 but there are no MCD model electrons at these altitudes. The neutral mass and argon density variations correlate with the electron density variations. This is noteworthy because these altitudes are where diffusive transport dominates, being well above those of the photochemically controlled region. The electrons and neutrals are reacting in concert with no direct coupling, i.e., low collision rates. This suggests vertical transport from regions where they do efficiently couple. In this example, the tidal variations in the ionosphere maintain correlation with the neutral density variation above the photochemical region. The normalized wave amplitudes of the electron densities are also about twice as large as those for the neutral mass density.

To show an example where tidal waves in the IT system with strong correlation between the electron and neutral density variations are identified at lower altitudes, closer to the peak electron density, in Figure 5 we present tidal variability in longitude of the electron and neutral densities, from both MAVEN observed and MCD model data, for 17 October through 5 November 2017 for altitudes 125 km, 135 km, and 145 km. This interval includes data from the eighth deep-dip campaign, which took place from 16–23 October 2017. During this deep-dip, MAVEN's periapsis reached ~125 km. At the altitude ranges shown, photochemical equilibrium should dominate. Note that a strong correlation between the electron density modulations and those in the neutral atmosphere is present for these altitudes. This behavior is what we would expect for the conditions of photochemical equilibrium at altitudes above the peak electron density; which should occur near this altitude range.

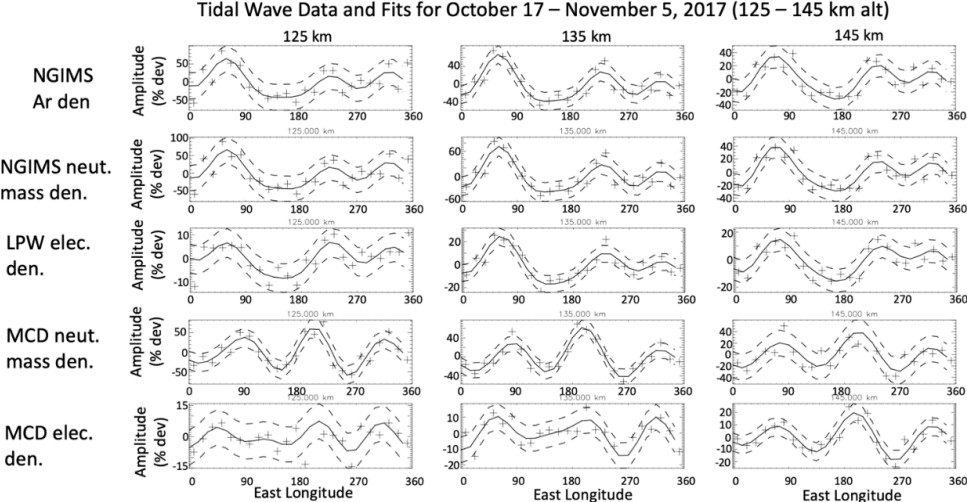

**Figure 5.** Tidal waves and corresponding fits (from the top row down) for the neutral argon density, neutral mass density, electron density, and MCD model density, for 17 October 5 November, 2017 for low altitudes, 125 km, 135 km, and 145 km, including those from the October 2017 deep dip. SLT ~14–12 h, SZA ~20°, latitude ~6–20° north.

The October 2017 deep dip described in Figure 5 is near subsolar. At altitudes of 125–145 km there is a clear strong positive correlation between the measured electron tidal perturbation and those in the neutral density. At higher altitudes (not shown) during this interval a strong correlation is not present. In the MCD model data (Figure 5, bottom two rows) the electrons and neutrals are positively correlated with one another, as is especially clear at 145 km. Comparing the observed data with MCD model results we see both are wave-3 dominated, but while the MAVEN-observed waveforms correlate with those in the MCD model within certain longitude ranges, at other longitudes they are out of phase. The amplitudes observed by MAVEN and the model wave amplitudes are comparable.

## 4. Statistical Study

To evaluate the IT tidal correlation on a statistical level, Figure 6 presents the correlation coefficients between the tidal wave fits for the electron and the neutral mass density variation (Figure 6a) and between electron and argon number density variation (Figure 6b), as a function of time and altitude. Note that the pattern in altitude of the correlation coefficients between electron and neutral mass densities and between electron and argon densities look very similar to one another, where we have both argon and neutral mass density observations, as is seen by comparing Figure 6a,b. The difference in Figure 6a,b is coverage in altitude, arising from the ability of NGIMS to routinely determine argon density at higher altitudes. We include both here because the argon densities give an idea of the higher altitude variability of the neutral atmosphere.

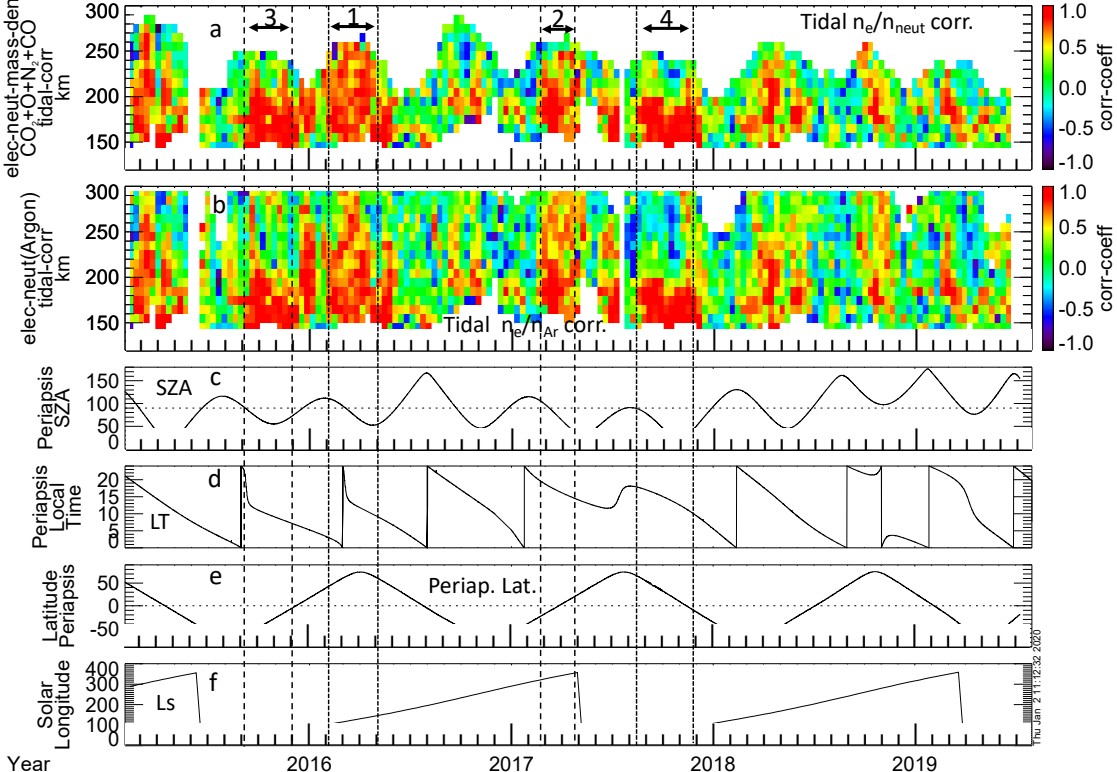

**Figure 6.** Correlation coefficients between the fits of the tidal variations in the electron density and neutral mass density (**a**), and electron density and argon density (**b**), as a function of altitude and time for most of the MAVEN mission; February 2015–July 2019. Correlation coefficients greater than 0.41 or anti-correlations less than −0.41 are statistically significant. The solar zenith angle (**c**), local time (**d**), latitude (**e**), at periapsis, and solar longitude (**f**) are shown for context. The vertical pairs of lines along with the numbers at the top of panel (**a**) mark the four intervals which are described in Section 4.

Periods and altitudes where the electron and neutral tidal variations have strong correlation (>0.5) appear orange to red on the color scale used. As determined using a t-test, correlations greater

than 0.41 or anti-correlations less than −0.41 are statistically significant at a 95% confidence level. There are intervals during which strong correlation occurs between electron and neutral density tidal variations at altitudes > 200 km. This is noteworthy because it is higher than the altitudes where direct (collisional) coupling occurs. One such interval, marked by "1" in Figure 6, occurs from early February to May 2016, $L_S$ from ~104° to ~147° (northern summer) a subinterval of which is represented in the data in Figures 2–4. This interval comprises a wide range of local times from post-midnight, dusk, and pre-noon, north latitudes between ~42–74°, SZA ~110–55°. A second such prominent interval with high altitude correlations ("2" in Figure 6) occurs from late February to late April 2017, $L_S$ ~320–357° (southern summer). The location for this second interval is latitude ~26° south–25° north, SZA ~109–34°, SLT ~19–14. Several other shorter intervals during which strong IT tidal correlations occur at altitudes > 200 km are also evident from Figure 6; these intervals tend to occur post-noon to pre-midnight (though not exclusively), as are the longer intervals just described. Hence, this high-altitude correlation does not seem directly dependent on location of SLT (except that it is a mainly a post-noon dayside phenomenon).

In contrast, there are also intervals in which the positive correlation is confined mainly to the photo-controlled altitudes. Two periods of strong correlation (correlation coefficient > ~0.6) from ~130–150 km (depending on the lower coverage limit) extending to ~200–210 km are present; the first ("3" in Figure 6) from early September to early December 2015, $L_S$ ~ 40–74°, and the second ("4" in Figure 6) from mid-August to late November 2017, $L_S$ ~51–97°. Both intervals occur at similar solar longitudes. In both of these intervals, the periapsis is located primarily on the dayside (SZA ranging from ~56–80° and ~20–87° respectively), mainly post-noon through noon to pre-noon (SLT 15–7 h) in the first case and from dusk through noon to pre-noon (SLT 20–11 h) in the second. The first interval occurs in the southern hemisphere, ~56° south to ~10° south, and the second primarily in the northern, ~63° north to ~10° south. The latter interval (namely mid-August to late November 2017) includes the deep dip interval shown in Figure 5.

To show the occurrence rate of statistically significant (positive) correlations between the tidal perturbations in electron and neutral densities, the percentage of time this is observed, per altitude, is given in Figure 7 for both the day (7–19 h SLT) and night sides (19–5 h SLT) separately, the extra hour of SLT included in the dusk side of the dayside range is to include regions of the atmosphere illuminated above the surface. The local times of 5–7 have been removed as a precaution because of the complex nature of the dawn terminator region [16,23]; though we find including this region does not significantly change the results. No distinction in the sorting has been made in $L_S$ or latitude. The IT coupling indicated by the correlation between electron and neutral density tidal variation is strongest on the dayside between 155 and 185 km.

Figure 8 shows the mean correlation of the electron tidal variations with the neutral mass and argon densities as a function of altitude for the night and daysides. The error bars are the standard deviation of the mean. No distinction in the sorting has been made in $L_S$ or latitude. Figure 8 shows clearly that stronger correlation occurs on the dayside. Specifically, the strongest correlation occurs on the dayside between 150 and 200 km with the peak correlation between the electron and neutral mass, and between electron and argon density, occurring at ~155 km, near the region where photochemistry should be dominant. This is consistent with findings by Mayyasi et al. [16] that identified strong correlations in the IT system at altitudes 160–200 km. On the nightside, a local peak in correlation between electrons and both argon and neutral mass peaks occurs at ~185 km with an average coefficient there of ~0.35; which means, on average, the correlation between the electrons and neutrals is below the level of statistical significance. However, as was shown in Figure 7, there does occur periods of statistically significant correlation in the night side about 40% of the time. That there are periods of IT correlation on the nightside suggest the presence of an ionizing mechanism. Fillingim et al. [25] demonstrated that accelerated electron precipitating into the night side Martian ionosphere leads to localized enchantments of the electron density and Fowler et al. [26] have shown that electron densities below 200 km can be maintained on the nightside by precipitating electrons.

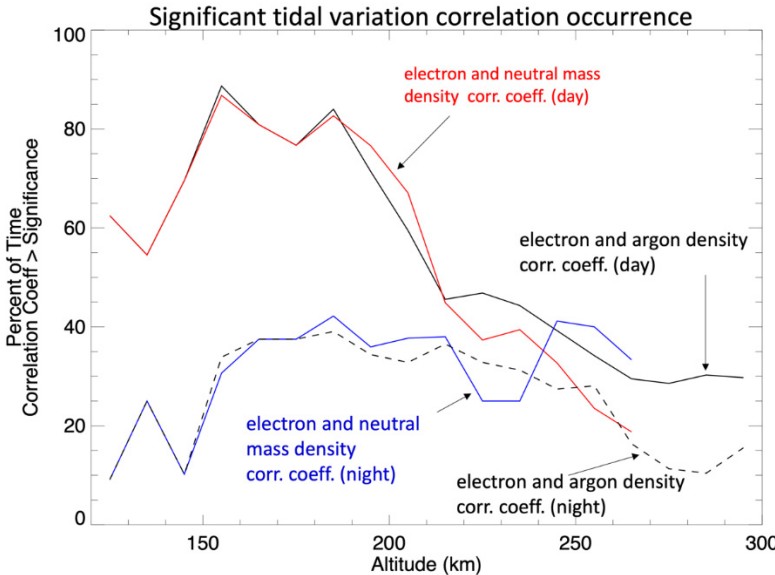

**Figure 7.** The percent occurrence of statistically significant correlations (>0.4) between the electrons and neutral tidal density variations as function of altitude for both day side (red and solid black) and night side (blue and dashed black).

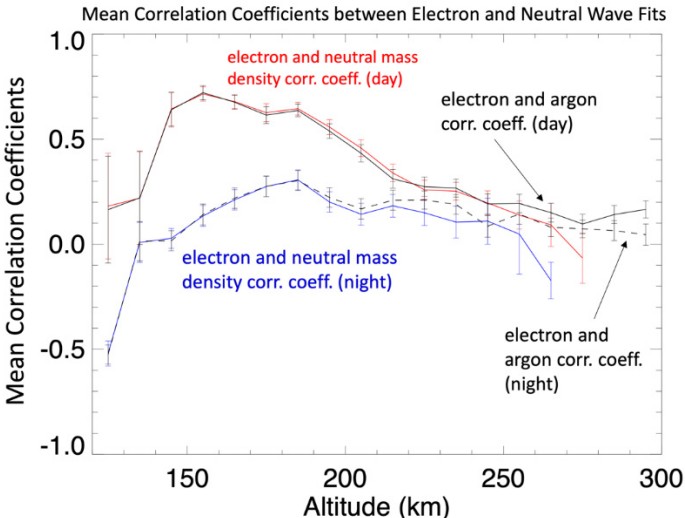

**Figure 8.** Mean values of the correlation coefficients between the electron tidal variations and neutral mass density (red on dayside, blue on nightside) and the argon density (solid black dayside and dashed nightside) as a function of altitude. The error bars are the standard deviation of the mean.

In order to compare the average amplitudes of neutral and electron tidal perturbations in altitude, we present the mean normalized tidal wave amplitudes on the night (blue) and dayside (red) as a function of altitude in Figure 9. Figure 9a shows the amplitudes for neutral argon number density (dashed) and neutral mass density (solid) perturbations, 9b shows the amplitudes of the modeled (MCD) neutral mass density perturbations, and 9c shows the amplitude of the MAVEN electron density perturbations.

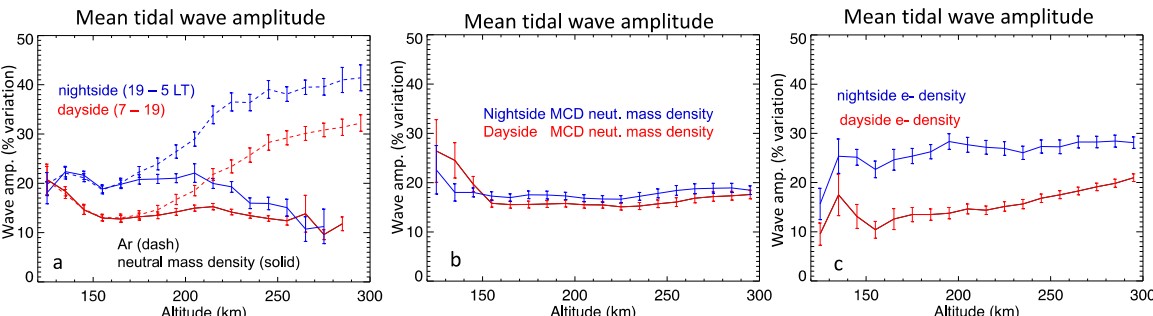

**Figure 9.** Mean normalized tidal wave amplitude as function of altitude for the NGIMS measured neutral atmosphere mass and argon densities (**a**), modeled neutral atmosphere mass density (**b**) and LPW measured electron density (**c**). The error bars are the standard deviation of the mean.

The tidal waves in both the neutral thermospheric density and ionospheric electron density have larger normalized amplitudes on the night side compared to the dayside. While both the neutral argon and mass densities have very similar average wave amplitudes below about ~200 km, above that altitude the argon tidal wave amplitudes increase with altitude. At 295 km, the average dayside (nightside) argon tidal wave amplitudes are 32% (41%) of their respective average background density. By contrast, above ~200 km the neutral mass density remains around ~10–15% (~10–20%) for dayside (nightside) amplitudes.

Larger amplitudes in the night side density perturbations are consistent with the behavior predicted by the barometric equation for an isothermal atmosphere; for lower temperatures (T) (i.e., nightside) the same relative temperature perturbation, $\delta T/T$, will produce larger relative density (n) perturbations, $\delta n_n/n_n$, because of an additional factor of $1/T$ in the full expression, see Appendix B Equation (A9). From Equation (A9) it is also evident that heavier species will be preferentially enhanced, given that $H = kT/mg$. This dependence has been demonstrated to occur in Mars tides by England et al. [6]. At altitudes below ~200 km, $CO_2$ is the dominant neutral constituent, and with a mass of 44 g/mol it is similar to the mass of Ar which is 40 g/mol, thus the normalized tidal perturbations should be similar. At higher altitudes, oxygen is the dominant constituent, the ratio between its mass (16 g/mol) and argon's mass is ~0.4. The ratio between the normalized neutral mass and argon perturbations at 250 km altitude is ~0.42 (~0.39) for the dayside (nightside).

On the dayside, the electrons tidal amplitudes are also ~10–20%; on the nightside the average amplitude of electron tidal waves are ~15–28%. These normalized variation amplitudes are similar to those of the neutral mass density. If the dependency of the electron variations on those of the neutrals is approximated from Equation (A7) we find the relation given in Equation (A8) which indicates that in the photochemically controlled region, at altitudes where the optical depth becomes small, the relative electron density perturbations are half those of the neutral densities. This holds assuming the relative perturbations are not too large (i.e., <40%); but larger perturbations in the normalized neutral density should result in the normalized electron density perturbations being less than a factor of $\frac{1}{2}$ that of the neutrals. The fact that the normalized perturbation amplitudes are similar, $\delta n_e/n_e \sim \delta n_n/n_n$, suggests that diffusion or some other processes of vertical transport is also involved. Even at altitudes <200 km this ratio is close to unity. At the lowest altitude, 125 km, the dayside variations follow the expected $\delta n_e/n_e \sim 0.5\ \delta n_n/n_n$.

The MCD model (Figure 9b) yields tidal wave amplitudes in the neutral mass density of ~16–17% (at > 150 km), similar to the amplitudes of the dayside observations. In the MCD model results, while the nightside amplitudes are larger than those on the dayside, the difference between these two amplitudes is much smaller than is observed in the MAVEN data.

To evaluate the dominant wavenumber of the identified tides, Figure 10 shows the dominant wavenumbers of the electron density variations (Figure 10a), neutral mass density variations (Figure 10b), argon densities (Figure 10c), and MCD neutral mass density variations (Figure 10d), determined from the wave analysis described above. Consistent with previous studies [6,12,24,27] the dominant neutral modes in both MAVEN and MCD results are very often wave-2 and wave-3; though wave-1 and wave-4 modes also have a significant presence at times. In the neutral atmosphere, wave-1 is overall statistically somewhat more prevalent than wave-3; wave-2 is the most frequently observed. We summarize the observation occurrence rate of these wave types in Table 1. The wave occurrence statistics for the neutral mass density (not shown) are very similar to the result obtained using argon.

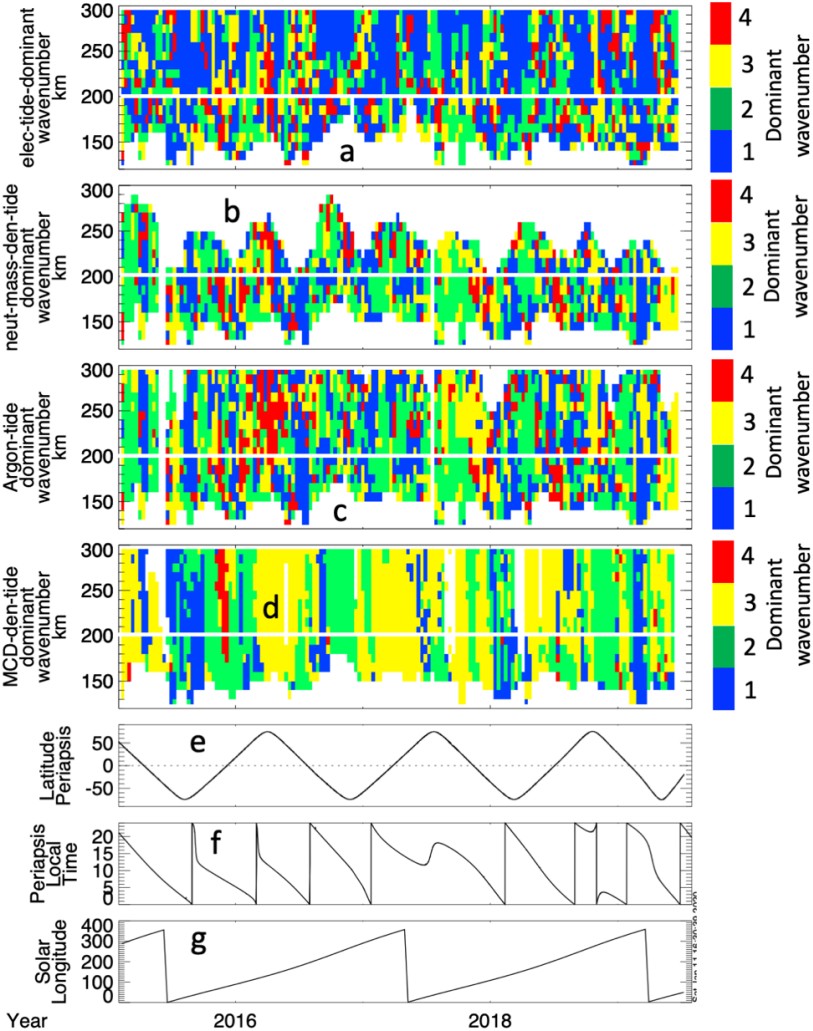

**Figure 10.** The dominant wave number, as determined from the fourth order harmonic fit, as a function of altitude and time from February 2015–July 2019 for the tidal variations observed in electron density (**a**), neutral mass density (**b**), neutral argon density (**c**), and modeled (MCD) neutral mass density (**d**). The local time (**e**), solar longitude (**f**), and latitude (**g**), are given for context. The white horizontal line marks 200 km.

**Table 1.** The fractional occurrence of observed dominant tidal wave modes in the MCD neutral mass densities, measured argon densities, and measured electron densities at high (>200 km) and low (<200 km) altitudes.

| Type | Altitude | Wave-1 | Wave-2 | Wave-3 | Wave-4 |
|---|---|---|---|---|---|
| Neutral (MCD) | >200 km | 16% | 33% | 49% | 2% |
| Neutral (MCD) | <200 km | 22% | 35% | 41% | 2% |
| Argon (NGIMS) | >200 km | 29% | 32% | 26% | 13% |
| Argon (NGIMS) | <200 km | 30% | 36% | 23% | 11% |
| Electron (LPW) | >200 km | 53% | 20% | 16% | 11% |
| Electron (LPW) | <200 km | 37% | 28% | 22% | 13% |

Because the magnetized electrons (which occur > 200 km), have higher density over regions of strong crustal fields [28] the wave-1 component of the electron tidal waves may be expected to be phased so that (in the south hemisphere at altitudes > 200 km) the peak in density is located over the longitudes of strong crustal fields, thus crustal fields may play a role organizing the wave pattern there. This is supported by closer inspection of the phase of the wave-1 electron tidal wave fits, shown in Figure 11, which are often phased such that the crest in electron density in the wave-1, above ~200 km, corresponds to the longitudes of strong crustal fields. The pairs of vertical lines delimit intervals where (above~200 km) the phasing of the wave -1 component puts the crest in electron density over the longitude range of southern hemisphere crustal fields. The color bar in 11a is adjusted so that the green regions (~150–250° longitude) roughly correspond to the longitudinal region where strong crustal fields are located (~140–230°). In 11b, between the pairs of vertical lines above ~200 km, it can be seen that wave -1 is often the dominant mode. This suggests that the crustal fields influence the ionospheric tidal perturbations at high (>200 km) altitudes, and may explain why, as shown in Table 1, the electron tidal perturbations are dominated by wave-1 53% of the time at high altitudes, compared to 37% at lower altitudes.

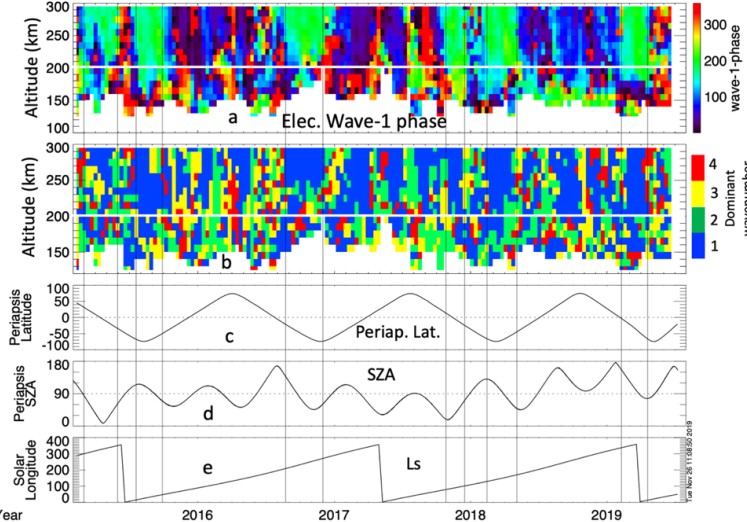

**Figure 11.** Panel (**a**) shows the phase of the wave-1 fit component electron perturbation indicating where the first wave crest is located (the only crest in the case of wave-1). Panel (**b**) shows the dominant wave number for the electron tidal harmonic fits (same as in Figure 10a). The latitude (**c**), solar zenith angle (**d**), and solar longitude (**e**) are shown for context.

To evaluate the phasing of the identified tidal wave variations in the neutral mass density as observed by MAVEN with those from the MCD, Figure 12 shows the correlation coefficients between the tidal variations in the MCD model neutral mass density and the neutral mass density determined

from the NGIMS measurements in panel (a). Panel (b) shows the correlation coefficient between the tidal variation of MCD neutral mass density and the argon density measured by NGIMS. There are intervals where the model and observations strongly correlate and others where they either have weak/no correlation or anti-correlate. The correlation tends to be the strongest in and around the evening sector.

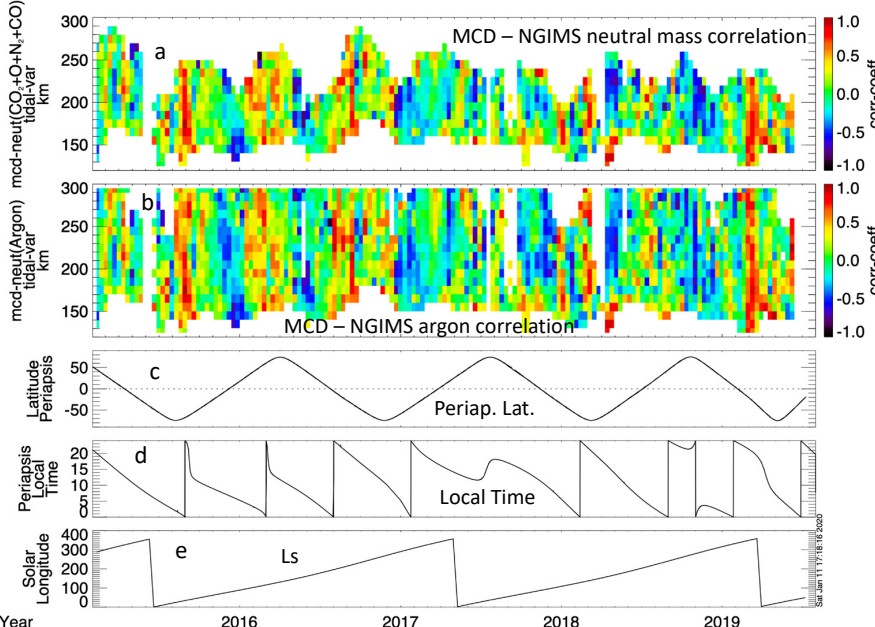

**Figure 12.** The correlation coefficients between the tidal waves identified in the MAVEN NGIMS-measured neutral mass density and the MCD model neutral mass density (**a**) as well as those identified in the MAVEN neutral argon density and MCD neutral mass density (**b**). The periapsis latitude (**c**), local time (**d**), and solar longitude (**e**) are shown for context.

A more quantitative comparison between the MCD and MAVEN neutral atmospheric tidal correlations is presented in Figure 13. Figure 13 shows the occurrence rate of (positive) statistically significant correlation between the tidal wave perturbation in MAVEN argon and neutral mass densities with MCD neutral mass density on both the day and night sides.

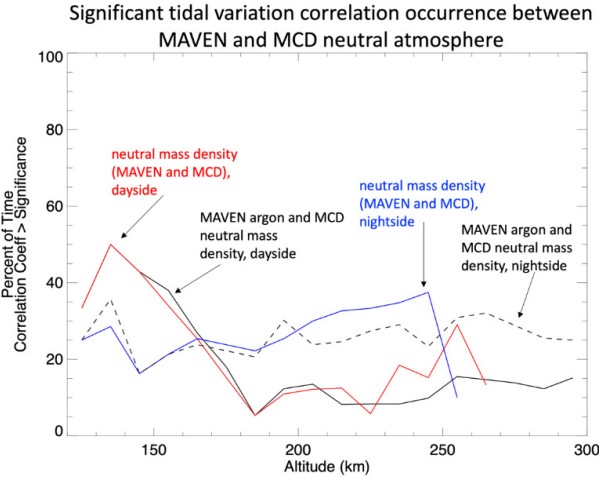

**Figure 13.** The occurrence rate of statistically significant positive correlations between the tidal wave perturbation identified in MAVEN neutral argon and mass densities with those identifies in MCD neutral mass density on both the day and night sides.

At altitudes below ~165 km, the tidal waves in the MAVEN and MCD neutral density have the largest correlation occurrence rate on the dayside; peaking at ~50% at 135 km. Above 165 km, the significant correlation occurrence on the nightside is ~30%.

## 5. Discussion

As was presented above, there are times when the tidal wave perturbations in ionospheric electron density strongly correlate with those in the thermospheric neutral density at altitudes above the region where photochemistry dominates (~>200 km). Intervals with these strong high-altitude correlations also show strong correlation at the lower, photochemically controlled altitudes. In addition, there are intervals during which the strong correlation between the electron and neutral tidal perturbations only occur at photochemically controlled regions.

The longer (>~2 months) intervals with strong correlation between the electron and neutral density, occur mainly on the dayside, as expected, given that that is where photochemistry will occur. This includes the intervals with strong correlation at high-altitudes (>200 km) above the photochemically controlled region. The fact that the occurrence of strong correlation at high altitudes is accompanied by those at lower altitudes suggests that photochemical processes are important even for the IT tidal correlations at altitudes where diffusive processes happen fast enough to dominate the dynamics. Diffusive processes result in upwards transport of plasma starting from the upper part of the photochemically controlled region. Other dynamical processes, such as interactions between the ionospheric plasma and either crustal magnetic fields or the magnetic pileup associated with the induced magnetic field may compete with diffusion, and could alter the ionospheric tidal wave behavior at higher altitudes causing it to behave differently than the thermospheric tides.

Above the photochemically controlled region, ion-neutral collisions and electron-ion dissociative recombination rates are low, and the ionosphere should not be directly locally coupled to the neutral thermosphere (or only weakly coupled). Since the density of the upward diffusively transported plasma should be, in part, controlled by the plasma density of the upper part of the photochemical region, variations in the thermosphere of the photochemical region could drive corresponding variations in the electron density, and the signature of these variations is propagated vertically upwards via diffusive transport. If no other plasma processes interfere, and the thermospheric density variations are mostly uniform phase-wise, in altitude from the top of the photochemical region upwards, then the correlation between the tidal variations in electrons and neutral should occur at the higher altitudes.

If the density perturbations observed were due to a purely photochemical behavior, then according to simplified photochemical theory the electron density perturbations normalized to background should be a factor of $\frac{1}{2}$ that of the neutral density perturbations (Equation (A8)). The fact that $\delta n_e/n_e \sim \delta n_n/n_n$ suggests that diffusion or some other processes of vertical transport is also involved.

One way to evaluate the relative roles of both photochemical and diffusive processes is to compare their characteristic timescales. A smaller timescale means that the corresponding process happen faster, and thus is more dominant in the system dynamics. We use Equations (A3)–(A6), (A9), and (A10) in Appendix B to estimate the timescales of photochemical, diffusion, and vertical motion of the neutral atmosphere for periapsis passes at ~9:44 UT on 8 April 2016 in Figure 14a, and at ~15:00 UT on 22 October 2017 (during a deep dip event) (Figure 14b) (from the intervals shown in Figures 2–5). The electron temperatures and densities where measured by the LPW Langmuir probe, neutral density by NGIMS, and the neutral temperature determined using the method described in Stone et al. [22]. For the ion temperature we have used a range of estimates comprising the electron temperature, neutral temperature, 300 K, 500 K, 700 K, and 1000 K; the actual ion temperature is expected to be someplace in between the neutral and electron temperature [29]. Full ion temperature profiles are not yet available due to ongoing calibrations, but by-eye analysis of this orbit performed by the STATIC instrument team suggests that ion temperatures approach 300 K at periapsis [K.G. Hanley, private communication]. While previous estimates of the altitude at which the diffusive timescale becomes smaller than the photochemical time scale are around ~200 km [5,13], for these events we estimate

somewhat lower altitudes, ~155–165 km, depending on the actual ion temperature. This is a typical altitude range for this analysis based on looking at other orbits. By altitudes of 200 km, the diffusive timescale is an order of magnitude smaller than the photochemical timescale, and thus the dynamics at these altitudes should be strongly dominated by diffusion.

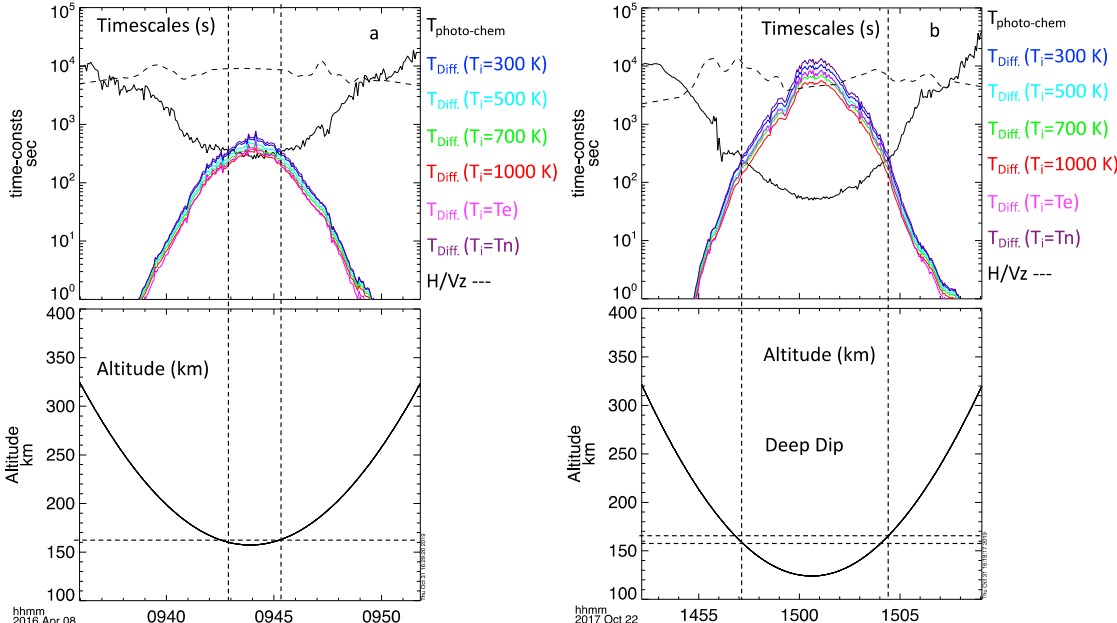

**Figure 14.** Calculations of the photochemical, diffusion, and neutral vertical motion time scales ~ +/- 300 km of periapsis at ~9:44 UT on 8 April 2016 (**a**) and ~15:00 UT on 22 October 2017 (**b**); times from the intervals shown in Figures 2–5 respectively. In the top panels, the black dashed trace is the timescale of vertical motion of the neutral atmosphere; the solid black trace is the photochemical timescale; the solid colored traces are the diffusive timescales for several possible ion temperatures indicated to the right of the panels. The bottom panels in both A and B show the MAVEN altitude. The horizontal dashed lines indicate the altitude, or range of altitudes, at which the photochemical and diffusion timescales are equal.

The photochemical timescale is inversely proportional to the electron density, while the diffusion timescale is proportional to neutral density, both densities decrease exponentially with altitude. Thus, the photochemical timescale increases exponentially with altitude while the diffusion timescale decreases exponentially with altitude. Hence a rapid cross over in the timescales; the diffusive timescale is not as sensitive to ion temperature as it is density, as is illustrated by the various diffusive timescale traces for different ion temperatures.

While prior investigations [1,4,14] have noted the consistency of tidal dynamics in IT coupling with photochemical processes, we show here that strong correlations also occur between electron and neutral densities above the photochemically controlled altitudes where diffusive processes should dominate. Figures 6 and 7 indicate that the ionospheric and tidal variations can be strongly correlated up to altitudes of ~265 km. At 265 km the diffusive timescale should be over three orders of magnitude smaller than the photochemical timescale. According to our estimate, the diffusive processes should play an important role in the IT coupling above 160 km. The conclusion that there are strong IT tidal correlations well above the photochemical region holds even if we assume the standard altitude of ~200 km for the timescales to crossover.

We also consider the relative vertical motion of the neutral atmosphere. To do this, the timescale associated with vertical motion of the neutral thermosphere as it is heated/cooled is calculated from Equation (A12) and plotted in Figure 14. The timescales of vertical motion of the neutral atmosphere are long relative to the photochemical timescales at altitudes < 200 km for the two intervals shown,

and the two timescales only become comparable at altitudes where the diffusive transport timescale is much smaller than either. Thus, as a parcel of atmosphere is lifted or falls within photochemically controlled altitudes, the photochemistry maintains equilibrium.

Based on our estimates of the diffusive and photochemical timescales, the two are similar around ~160 km, the ionospheric dynamics there should be controlled by a combination of both photochemistry and diffusion. This supports the idea that the reason the relative fractional amplitudes of the electron tidal wave perturbations are comparable to those of the neutral densities, as opposed to being a factor of $\frac{1}{2}$ that of the neutral density perturbations, is that the dominant process is not photochemistry by itself, but a combination of the two.

## 6. Conclusions

Of the various processes that give rise to electron density variability in the Mars ionosphere, one important source is tidal waves. Here we verify the close coupling of the IT system for altitudes <200 km, and demonstrate that there is a close correspondence in the IT system at higher altitudes, where the neutrals and ions are not linked by collisions or simplified photochemical-type physics. We have done this by analyzing tidal wave perturbations in over two Martian years of MAVEN in situ measurement of ionospheric electron density and neutral atmospheric density. The correlation between the electron and neutral density tidal wave perturbations, and the amplitudes of these waves were examined. The results of the analysis show that in addition to the theoretical expected behavior of the IT coupling at photochemically controlled altitudes, there are periods where strong correlations between the ionospheric and thermospheric density perturbation extend to higher altitudes (>~200 km). Prior investigations [1,4,14] have shown the consistency of tidal dynamics in the ionosphere as being coupled to the neutral variations through photochemical processes. In this paper, we show that strong correlations between electron and neutral densities above the photochemically controlled region, at altitudes where diffusive processes dominate, also occur in tidal dynamics. The positive correlation between the neutral and electron densities in the photochemical regime (but where the optical depth is less than unity) can be explained by simplified photochemical theory. In the diffusive region, plasma processes play an important role in the behavior of the relation between neutral and electron densities. The likely explanation for the periods of strong correlation at high altitudes is that the thermospheric density variation maintain approximately constant phase in altitude from the top of the photochemically controlled region on higher. The top of the photochemically controlled region supplies the higher altitude regions with much of its electron population, as is evident from the studies showing actual electron densities higher than what can be explained through local ionization (i.e., photochemical physics) [30]. Since in the upper part of the photochemically controlled region the electrons and neutrals should vary in phase, the variations in the electron density can be projected upwards by the diffusively transported electrons. Other intervening plasma processes, such as induced magnetic field structures like the magnetic pile up boundary, or crustal magnetic fields may disrupt this correlation. In the southern hemisphere, we have noted the electron density perturbations often exhibit dominant wave-1 structure, with phasing such that the higher density region corresponds to the longitudinal region where crustal fields are the strongest.

To evaluate whether the current day Mars model (MCD) can capture neutral tidal waves in the thermosphere, we compared the MAVEN data set to the MCD model that included high resolution and large-scale perturbations. Comparison between the tidal wave amplitudes in the measured neutral densities and model (MCD) neutral densities show a reasonable agreement between them, especially on the dayside. While the phasing of the model and observed tidal waves sometime agrees well (strong correlation), there are other times where there is either no correlation or anti-correlation. Significant correlation between the MAVEN and MCD neutral tidal variations occurs on average with greater frequency on the dayside below ~165 km, peaking at ~50% around 135 km. Above 165 km the occurrence of correlation is greater in the night side, being present about ~25% of the time. Thus, the modeled thermosphere seems to be able to reasonably capture the amplitude well, but phase/wave

number often differs. We also have compared the MCD tidal waveforms at 195 km to those at 275 km and found that the phase changes little, ±~15° on average in this altitude range. This shows that the modeled tidal wave phase changes little in the region above the photochemically controlled region. The tidal perturbations in the model neutral density typically correlate strongly with those in the model electron densities. But since the MCD electrons are available only in the photochemistry-controlled region, the impact of the thermosphere on the ionosphere at high altitudes cannot yet be fully studied with the model.

**Author Contributions:** Conceptualization, S.A.T., L.A., M.D.P. and E.T.; methodology, S.T., L.A., S.B., E.T., P.W., M.E. and M.D.P.; software, S.A.T., L.A., P.W. and M.D.P.; validation, S.A.T., L.A., X.F., E.T. and M.E.; formal analysis, S.A.T. and L.A.; investigation, S.A.T., L.A., M.D.P., E.T., M.E., P.W., S.B., X.F. and F.G.-G.; data curation, S.A.T., L.A., M.D.P., M.E. and F.G.-G.; writing—original draft preparation, S.A.T. and L.A.; writing—review and editing, S.A.T., L.A., E.T., X.F., P.W., S.B., G.J. and M.D.P.; visualization, S.A.T., L.A., X.F., P.W., M.E., S.B. and M.P. All authors have read and agreed to the published version of the manuscript.

**Funding:** The MAVEN project is supported by NASA through the Mars Exploration Program. Work at the Laboratory for Atmospheric and Space Physics, at Boston University and at the University of Michigan was done under the MAVEN project. F.G. is funded by the Spanish Ministerio de Ciencia, Innovación y Universidades, the Agencia Estatal de Investigacion and EC FEDER funds under project RTI2018-100920-J-I00, and acknowledges financial support from the State Agency for Research of the Spanish MCIU through the "Center of Excellence Severo Ochoa" award to the Instituto de Astrofísica de Andalucía (SEV-2017-0709). X.F. is funded by NASA grant 80NSSC19K0562.

**Conflicts of Interest:** The authors declare no conflicts of interest.

## Abbreviations

The following abbreviations are used in this manuscript:

| | |
|---|---|
| IT | Ionosphere-thermosphere |
| LPW | Langmuir Probe and Waves |
| Ls | Solar Longitude |
| LST | Local Solar Time |
| MAVEN | Mars Atmosphere and Volatile Evolution |
| MCD | Mars Climate Database |
| NGIMS | Neutral Gas and Ion Mass Spectrometer |
| STATIC | SupraThermal And Thermal Ion Composition |
| SZA | Solar Zenith Angle |

## Appendix A. Tides

Solar thermal tides on Mars are planet-scale oscillations of atmospheric density, pressure, temperature, and wind velocities, on timescales of a Martian day (1 sol) and subharmonics, which result from solar heating and longitudinal/zonal asymmetries associated with surface topography [7–9,12]. The standard way of mathematically representing the amplitude of the tidal oscillations as a function of universal time, t, and longitude, $\lambda$, [8,24] is given by

$$A_{n,s} \cos\left(n\Omega t + s\lambda - \phi_{n,s}\right) \tag{A1}$$

where, $A_{n,s}$, is the amplitude, $\Omega$ is Mars' angular rotation frequency ($\Omega = 2\pi \times \text{sol}^{-1}$), n is a subharmonic of the sol (=1,2, … ), s is the zonal wavenumber (s = … ,−2,−1,0,1,2, … ) with s < 0 corresponding to eastward propagating waves and s > 0 to westward propagating, and $\phi_{n,s}$ is the phase. Rewriting (A1) as a function of solar local time, $t_{LT}$, where the relation between the solar local time and universal time is given by

$$t_{LT} = t + \frac{\lambda}{\Omega} \tag{A2}$$

the expression (A1) becomes:

$$A_{n,s} \cos\left(n\Omega t_{LT} + (s-n)\lambda - \phi_{n,s}\right) \tag{A3}$$

The total amplitude will be given by a summation of (A3) over all n and s. When n = s there is no longitudinal dependence, this case corresponds to a zonally symmetric atmosphere and surface [8]. Non-migrating tides are the case in which the wave is described by n ≠ s.

For non-migrating tides, an observer fixed in local time measures waves in longitude as the planet rotates underneath, bringing the different longitudes to any one local time where the observer happens to be. The apparent longitudinal structure in the observed tides is a result of the fixed local time measurement and the waves propagating eastward or westward with a diurnal (24 h, n = 1), semidiurnal (12 h, n = 2) of terdiurnal (8 h, n = 3) periods. Thus, a diurnal eastward propagating wave with zonal number 2 (DE2) is observed at a fixed local time as a wave 3; in general, |s − n| maxima or minima.

## Appendix B. Ionosphere Timescales

The photochemical timescale in the Mars ionosphere is primarily that of dissociative recombination of electrons with $O_2^+$ [5], and is dependent on the electron density, $n_e$, and the $O_2^+$ dissociative recombination rate, $\alpha$, and is given by:

$$\tau_{pc} = \frac{1}{\alpha n_e} \tag{A4}$$

The $O_2^+$ dissociative recombination rate depends on the electron temperature, $T_e$, and is given by [31];

$$\alpha = 1.95 \times 10^{-7} \left( \frac{300K}{T_e} \right)^{0.7} cm^3 s^{-1} \text{ for } T_e < 1200 \text{ K} \tag{A5}$$

$$\alpha = 7.38 \times 10^{-8} \left( \frac{1200K}{T_e} \right)^{0.56} cm^3 s^{-1} \text{ for } T_e > 1200K \tag{A6}$$

Under the conditions of one neutral species, isothermal atmosphere, and a constant ionization cross section in EUV wavelength, the electron density in the region governed by photochemical equilibrium, where photoionization and recombination are in equilibrium, is described by simplified photochemical theory, i.e., the Chapman profile [31], which can be written in terms of the (isothermal) neutral density, $n_n(z)$, at height z from the planetary surface, or other reference altitude, as

$$n_e = \sqrt{\left( \frac{\eta \sigma I_\infty}{\alpha} \right) n_n(z) e^{-\frac{1}{2} \sec(\chi) \sigma H n_n(z)}} \tag{A7}$$

where $\sigma$ is the ionization cross section, $\eta$ the ionization efficiency, $I_\infty$ the EUV flux at the top of the atmosphere, H the atmospheric scale height, and $\chi$ the solar zenith angle. A number of works have confirmed that the Chapman profile describes reasonably well the Martian ionospheric electron density from the altitude of peak electron density, ~130 km, up to between ~180 km and ~200 km using ionospheric sounding [5] and using in situ MAVEN LPW Langmuir probe measurements [30,32].

Taking the derivative of $n_e$ with respect to $n_n(z)$ in Equation (A7), and rewriting the resulting expression as the ratios of the change ($\delta$) in densities, (approximating by the differentials) to the respective absolute values it can be shown that

$$\frac{\delta n_e}{n_e} \approx \frac{1}{2} \frac{\delta n_n(z)}{n_n(z)} (1 - \tau) \tag{A8}$$

where $\tau$ is the optical depth, $\tau \equiv n_n(z) \sigma H \sec(\chi)$. It can be seen that for altitudes at which $\tau < 1$ the electron and neutral densities will vary in phase, provided the neutral atmosphere density changes slowly enough to maintain photochemical equilibrium. This expression assumes the relative perturbations in density are not too large (<40%). For the case of larger density perturbations, the ratio $n_e / n_e < \frac{1}{2} \delta n_n / n_n$.

If the variation in neutral density is brought about by temperature variations:

$$\frac{\delta n_n}{n_n} = \frac{\delta H}{H} \left( \frac{z}{H} - 1 \right) \tag{A9}$$

where z is the altitude from the surface or other reference altitude. Equation (A9) is obtained from writing the barometric equation with the reference density $n_n(z = 0)$ as a function of the scale height, taking the derivative of the density, $n_n$, with respect to H, and rewriting the differentials of density and scale height as fractions to the corresponding full values.

The diffusion time scale for dynamics governed by the balance between the plasma thermal pressure gradient and ion-neutral collisional force is given by

$$\tau_D = \frac{H^2}{D} \tag{A10}$$

where D is the plasma diffusion coefficient given by (8);

$$D = \frac{k_B (T_e + T_i)}{m_i \nu_{in}} \tag{A11}$$

where $k_B$ is Boltzmann constant, $T_i$ is the ion temperature, $m_i$ the ion mass, and $\nu_{in}$ the ion neutral collation frequency [5,13].

In the case of tidal motions, one can also consider the vertical motion the neutral atmosphere and whether the timescale associated with this motion is fast or slow relative to that of photochemical equilibrium.

Heating and cooling of the thermosphere result in vertical motion with a velocity, vz:

$$v_z = \frac{k_B}{mg} \frac{dT}{dt} \frac{z}{H} = \frac{dT}{dt} \frac{z}{T} \tag{A12}$$

This can be deduced as the rate of vertical displacement of a constant pressure surface in an isothermal atmosphere with uniform rate of temperature change. For Mars, at an altitude of 200 km, temperature of 230 K, and rate of temperature increase rate of 0.0016 K/s (estimates from examination of the variations in argon temperatures at 200 km) we obtain a velocity of ~1.4 m/s. The vertical motion timescale can be defined as $H/v_z$. A standard Mars atmospheric scale height of ~11 km yields a timescale of neutral vertical motion of ~7860 sec. At photochemical altitudes, where $\tau_{PC} \sim 100$ sec, $H/v_z >> \tau_{pc}$, so photochemical equilibrium is maintained as a parcel of atmosphere moves vertically described by Equation (A12).

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
