# Peer review of "Tidal Wave-Driven Variability in the Mars Ionosphere-Thermosphere System"

_atmosphere, doi:10.3390/atmos11050521_

Round 1

Reviewer 1 Report

Manuscript: Tidal wave driven variability in the Mars ionosphere-thermosphere system, by S.A. Thaller, L. Andersson, M. D. Pilinski, Edward Thiemann, Paul Withers, Meredith Elrod, Xiaohua Fang, F. González-Galindo, Stephen Bougher and Geoffrey Jenkins (Atmosphere manuscript ID atmosphere-797983)

Summary: MAVEN measurements in the Martian atmosphere show that correlations between electron and neutral atmosphere densities, previously observed at low altitudes (150-185 km), are sometimes observed at higher altitudes, up to 270 km. Electron densities at high altitudes show tidal variations similar to those seen at lower altitudes, where the thermosphere directly affects ionosphere densities through photochemistry. In addition, MAVEN data is compared to the pseudo-empirical MCD model, which is based on a database of Mars simulations. It is suggested that the correlation between electron and neutral densities at high altitudes is attributable to rapid upward diffusion of plasma from the top of the photochemistry region and a lack of variation in tidal phase versus altitude. The MCD model compares well to MAVEN, but MCD tends to shift in phase (versus longitude) relative to MAVEN.

Review: This manuscript includes both an analysis of MAVEN data and a check on the validity of the MCD model of the Mars climate. The presentation is clear and interesting. As is appropriate, the check on the validity of the MCD model is treated as a secondary issue. As described in the manuscript, the fact that correlations at low altitude persist up to high altitudes suggests that neutral tidal waves don’t change phase versus altitude in this region. Otherwise upward diffusion might phase-mix the variations, reducing them at high altitude. From the manuscript, it is not clear whether or not the authors checked to see if the MCD model reproduced this hypothesized constancy in the tidal phase versus altitude. MCD might have a problem wherein phase relationships present in a single simulation are clouded when multiple simulations are averaged as might happen in a model based on a database of many simulations. It would help if this central new physical result were checked against the model.

Additional minor comments:

Line 113: This reviewer would prefer that model outputs not be called “data.” To the degree possible, this term should be reserved for observations or other measurements of the physical universe. But this is just a preference on the part of the reviewer.

Lines 210-218: These lines seem to have a formatting issue (the software seems to be mistaking this paragraph for a figure caption).

Line 143: “To quantity how well various pairs of wave perturbations are phased…” Should this be “To quantify…”?

Lines 442-445: “The fact that the occurrence of strong correlation at high altitudes is accompanied by those at lower altitudes suggests that photochemical processes are important even for the IT tidal correlations at altitudes where diffusive processes happen fast enough to dominate the dynamics.”

This reviewer was unsure what was being conveyed by this sentence. After reading the rest of the manuscript, its meaning became clear. Perhaps it could be improved. What other dynamical processes compete with diffusion for dominance?

Line 616: There appears to be a formatting issue with Equation 6B.

Author Response

Thaller et al., Tidal wave driven variability in the Mars ionosphere-thermosphere system

Response to Reviewer 1

We would like to thank the reviewer for reading the manuscript and providing helpful suggestions for improving the manuscript.  We have incorporated all of the reviewer’s suggestions in the revised text.  Below, we have reproduced the reviewer’s suggestions in blue italicized text.  The reviewer’s suggestions are then followed by a description of how we modified the manuscript in response.  In the revised manuscript, all edits are indicated by yellow highlighting of the added or modified text.

“From the manuscript, it is not clear whether or not the authors checked to see if the MCD model reproduced this hypothesized constancy in the tidal phase versus altitude. MCD might have a problem wherein phase relationships present in a single simulation are clouded when multiple simulations are averaged as might happen in a model based on a database of many simulations. It would help if this central new physical result were checked against the model.”

We have checked to see if the MCD model shows a consistency of tidal phase in altitude and find that it typically does. For example, comparing the MCD tidal wave forms at 195 km to those at 275 km (to make a comparison of the phase near the top of the photochemical region to those at an altitude well above that), we find that on average the phase change is +/- ~15o.  We have added text to this effect on lines 566- 569 of the revised manuscript.

Additional minor comments:

Line 113: This reviewer would prefer that model outputs not be called “data.” To the degree possible, this term should be reserved for observations or other measurements of the physical universe. But this is just a preference on the part of the reviewer.

We have changed the text in several places (lines 88, 113, and 121) referring to “results” instead of “data”, or naming the modeled quantity directly (i.e. “neutral densities” instead of “data”).

Lines 210-218: These lines seem to have a formatting issue (the software seems to be mistaking this paragraph for a figure caption).

Yes, these lines are not supposed to be formatted as a figure caption.  We have changed the formatting to match that of the body of the main text. Lines 211 – 220.

Line 143: “To quantity how well various pairs of wave perturbations are phased…” Should this be “To quantify…”?

Yes, it should be “To quantify…”.; we have made this correction. Line 143.

Lines 442-445: “The fact that the occurrence of strong correlation at high altitudes is accompanied by those at lower altitudes suggests that photochemical processes are important even for the IT tidal correlations at altitudes where diffusive processes happen fast enough to dominate the dynamics.”

This reviewer was unsure what was being conveyed by this sentence. After reading the rest of the manuscript, its meaning became clear. Perhaps it could be improved. What other dynamical processes compete with diffusion for dominance?

To make the sentence cited clearer, we have followed the reviewer’s suggestion and added mention of other dynamical processes that may be competing with diffusion on lines 449 – 453.

Line 616: There appears to be a formatting issue with Equation 6B.

There does seem to be a formatting issue.  We have change “Equation 6B” to “(6B)”.  Line 630.

Reviewer 2 Report

Review of “Tidal wave driven variability in the Mars ionosphere-thermosphere system” by S.A. Thaller, L. Andersson, M.D. Pilinski, E. Thiemann, P. Withers, M. Elrod, X. Fang, F. Gonzalez-Galindo, S. Bougher, G. Jenkins

In this paper, the authors analyze over four years of MAVEN in-situ measurements of electrons and neutral densities in the Martian ionosphere/thermosphere. The authors show that there are strong correlations between tidal perturbations of the electrons and neutrals during the daytime at altitudes dominated by photochemistry. In addition, there are periods in which the correlations extend to higher altitudes, which are attributed to diffusive transport of plasma. Results are compared with the Mars Climate Database (MCD), which is derived from a GCM of the Martian atmosphere. This paper is clearly written and well-organized, but would benefit from some clarification. Below are specific comments which should be addressed before this manuscript is suitable for publication.

Line 72: “(LWP) [17] and the”

Line 117: Please include the version of MCD used in this study.

Line 158: There’s a word missing in this sentence between “the” and “gave”.

Line 174: Discussion of Figure 2. It is not that easy to see that the enhancements are specifically at 30, 150, and 290 degrees and that the crests shift 15 degrees westward. It would be worth mentioning that these features of the waves are easier to discern in Figure 3.

Line 185: The “+” symbols do not correspond to the bin colors, as stated in the text.

Line 200: Please make this sentence clearer. Which altitudes? And positive correlations between what?

Line 228: What are the dates of the deep-dip campaign in October 2017?

Figure 6: Perhaps include vertical dashed lines to delineate the various periods discussed in the text in Section 4. At the very least, make the tick marks easier to see on the time axis.

Line 481: This sentence is ambiguous; what is decreasing exponentially, density or timescale?

Line 486: Include references for “prior investigations”.

Figure 14: Subscript text is very small. Temperature units should be upper case K. Indicate which is figure (B) in the caption. Include a description of the bottom panels.

Author Response

Thaller et al., Tidal wave driven variability in the Mars ionosphere-thermosphere system

Response to Reviewer 2

We would like to thank the reviewer for reading the manuscript and providing helpful suggestions for improving the manuscript.  We have incorporated all of the reviewer’s suggestions in the revised text.  Below, we have reproduced the reviewer’s suggestions in blue italicized text.  The reviewer’s suggestions are then followed by a description of how we modified the manuscript in response.  In the revised manuscript, all edits are indicated by yellow highlighting of the added or modified text.

Line 72: “(LWP) [17] and the”

We added “and”.  Line 73.

Line 117: Please include the version of MCD used in this study.

Added “version 5.3”.  Lines 113 and 117.

Line 158: There’s a word missing in this sentence between “the” and “gave”.

The phrase “of the” in the original manuscript was deleted, resulting in the new sentence: “Experiments with calculating the mean RMS wave amplitude per altitude gave similar amplitudes and curve shape in amplitude-altitude space.” Lines 158 – 159.

Line 174: Discussion of Figure 2. It is not that easy to see that the enhancements are specifically at 30, 150, and 290 degrees and that the crests shift 15 degrees westward. It would be worth mentioning that these features of the waves are easier to discern in Figure 3.

We agree that these features are more discernable in Figure 3, and by comparison of Figure 3 (low altitudes) with Figure 4 (high altitudes) for the phase shift.  We have thus added “These features can also be seen by comparing the waveforms presented in Figure 3 with those in Figure 4” on lines 177 – 178.  

Line 185: The “+” symbols do not correspond to the bin colors, as stated in the text.

The way we originally worded that description was confusing. We have modified the text to be more succinct.  Line 187: “The ‘+’ symbols are the actual data, the solid black line….”.

Line 200: Please make this sentence clearer. Which altitudes? And positive correlations between what?

We have added more information to this sentence to make it clearer.  It now reads “At these altitudes (155 – 175 km), on the dayside, a positive correlation between electrons and neutral densities is consistent with tidal variation in the ionosphere driven by variation in the neutral atmosphere in photochemical equilibrium with the ionosphere.”  Line 201-203.

Line 228: What are the dates of the deep-dip campaign in October 2017?

The dates of the October 2017 deep-dip campaign are October 16 – 23, 2017.  We have added this information to the text on line 231: “This interval includes data from the eighth deep-dip campaign, which took place from October 16 – 23, 2017.”

Figure 6: Perhaps include vertical dashed lines to delineate the various periods discussed in the text in Section 4. At the very least, make the tick marks easier to see on the time axis.

We have both made the tick marks easier to see and added vertical lines to mark the specific intervals on Figure 6 described in section 4. We have also labeled these intervals 1 through 4 on Figure 6 and added references in the text to these intervals by number.  Lines 266, 270, 280, 281, 293,294.

Line 481: This sentence is ambiguous; what is decreasing exponentially, density or timescale?

We have modified the text to clarify that it is the densities that are exponentially decreasing with altitude; line 489 – 490.

Line 486: Include references for “prior investigations”.

References to “prior investigations” have been added.  Line 494.

Figure 14: Subscript text is very small. Temperature units should be upper case K. Indicate which is figure (B) in the caption. Include a description of the bottom panels.

We have increased the size of the text in the margin labels and used upper case K for temperature units.  Figure (B) has been indicated in the figure caption and we have added a description of the bottom panels. Lines: 516 – 518.
